# 🦉 FKA-Owl: Advancing Multimodal Fake News Detection through Knowledge-Augmented LVLMs

## ABSTRACT

The massive generation of multimodal fake news involving both text and images exhibits substantial distribution discrepancies, prompting the need for generalized detectors. However, the insulated nature of training restricts the capability of classical detectors to obtain open-world facts. While Large Vision-Language Models (LVLMs) have encoded rich world knowledge, they are not inherently tailored for combating fake news and struggle to comprehend local forgery details. In this paper, we propose FKA-Owl, a novel framework that leverages **f**orgery-specific **k**nowledge to **a**ugment LVLMs, enabling them to reason about manipulations effectively. The augmented forgery-specific knowledge includes semantic correlation between text and images, and artifact trace in image manipulation. To inject these two kinds of knowledge into the LVLM, we design two specialized modules to establish their representations, respectively. The encoded knowledge embeddings are then incorporated into LVLMs. Extensive experiments on the public benchmark demonstrate that FKA-Owl achieves superior cross-domain performance compared to previous methods. Code will be made publicly available.

## CCS CONCEPTS

• **Social and professional topics**; • **Security and privacy** → **Human and societal aspects of security and privacy**; **Social aspects of security and privacy**; *Usability in security and privacy*;

## KEYWORDS

Multimodal Fake News Detection, Large Vision-Language Model, Knowledge Augmentation

## 1 INTRODUCTION

The wide spread of fake news has become an important social issue, posing threats to politics [11], finance [10], and public health [36]. Misusing advanced generative models to create misinformation further exacerbates these issues, manifested in the rise of both text fake news [50] and also visual deepfakes [60]. Furthermore, multimodal forgery media through convergence disseminates more expansive information with greater impact to mislead readers. Detecting such fake news poses a unique challenge due to the existence of manipulations in both image and text modalities.

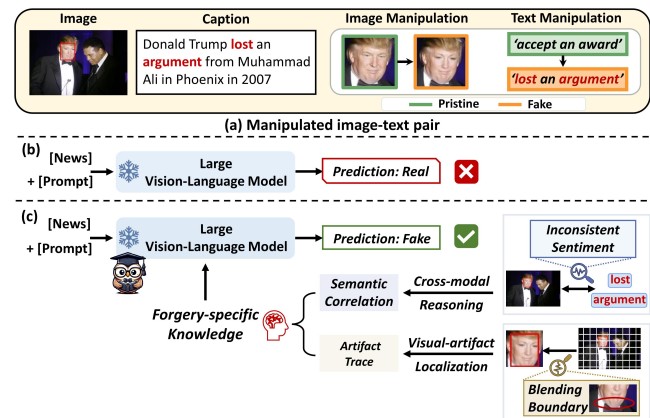

**Figure 1: Illustration of the effect of forgery-knowledge augmentation. (a) An example of a manipulated image-text pair in which Trump's face is swapped with another person and the positive words "accept an award" is replaced with the negative "lost an argument". (b) Existing LVLMs struggle to correctly judge the news veracity. (c) Incorporating forgery-specific knowledge (i.e., semantic correlation and artifact trace) into LVLM helps the model make accurate predictions.**

Faced with this challenge, current multimodal fake news detection (MFND) methods [42, 47, 57] primarily focus on incorporating cross-modal features. While some progress has been made, the acquisition of broad world information remains challenging due to the confinement of training to given domains (i.e., closed systems). However, open-world fake news exhibits substantial distribution discrepancies [63], termed domain shift [38, 64], which is manifested in abundant forgery methods and diverse real-world context. The existence of domain shift increases the difficulty of understanding and characterizing open-world fake news in MFND tasks.

To address this problem, we propose to leverage Large Vision-Language Models (LVLMs) [31, 62] which possess rich world knowledge for fake news detectors. This knowledge encompasses a wide array of world facts [4] about public celebrities, social events etc, which enables a comprehensive understanding of agnostic fake news. However, despite their proficiency in recognizing common instances, *the performance of applying off-the-shelf LVLMs to the MFND task is not always satisfying*. On the one hand, since LVLMs are not inherently tailored for MFND tasks, they are still challenging to understand and discover the subtle cross-modal differences [41]. For example, in detecting manipulated image-text pair in Fig. 1-(a), the model must discern the sentiment tendencies of both image and text, which are typically not present in LVLMs' training data. On the other hand, LVLMs lack sensitivity to localized spatial details [58]. As shown in Fig. 1-(a), when swapping Trump's face with

another person, there exists intrinsic image discrepancies between edited regions and pristine backgrounds, which is hard to perceive with existing LVLMs. Therefore, it is important to augment LVLMs with external information, referred to as *forgery-specific knowledge*, which is absent from model parameters yet useful for manipulation reasoning.

We identified two types of forgery-specific knowledge essential for manipulation reasoning [61]: *semantic correlation* and *artifact trace*. Firstly, manipulated media often disrupt the natural coherence between different modalities, resulting in semantic discrepancies [47]. Secondly, alterations in images usually produce distinctive artifact traces, such as irregular blending boundaries and inconsistencies in color sources, among others [51]. As a result, it would be beneficial to incorporate these two types of knowledge into the training and inference of large vision-language models for multimodal fake news detection.

In this paper, we present a novel framework, namely FKA-Owl which augments LVLMs with forgery-specific knowledge to enhance cross-domain performance for multimodal fake news detection. This framework leverages the rich world knowledge of LVLMs, supplementing it with domain-specific knowledge crucial for identifying multimodal fake news. As shown in Fig. 1-(c), to achieve that, we establish representations of the aforementioned two kinds of forgery-specific knowledge with two specialized modules. The cross-modal reasoning module applies dual cross-attention mechanisms to integrate visual and textual information from frozen encoders, aiming to identify semantic inconsistencies. Meanwhile, the visual-artifact localization module focuses on detecting precise visual artifacts at multiple levels of detail, using sparse bounding boxes and detailed mask regions to trace artifacts. Subsequently, the encoded knowledge representation embeddings are mapped to the language space of LVLMs with projectors. We devise MFND instruction-following data for fine-tuning and employ both candidate answer heuristics and soft prompts to unleash the extensive knowledge of language models.

Our contributions are summarized as follows:

- We pioneer leveraging rich world knowledge from large vision-language models (LVLMs) and enhancing them with forgery-specific knowledge, to tackle the domain shift issue in multimodal fake news detection. Our proposed method, FKA-Owl, serves as a general detector to bridge the gap.
- FKA-Owl augments LVLMs with forgery-specific knowledge for manipulation reasoning. We propose two lightweight modules: the cross-modal reasoning module and the visual-artifact localization module to extract semantic correlations and artifact traces, respectively.
- The extensive experiments demonstrate the effectiveness of our proposed method under multiple cross-domain settings.

## 2 RELATED WORK

### 2.1 Fake News Detection

Fake news detection works can be categorized into unimodal (image-based and text-based) and multimodal methods. Image-based method [5] proposes to exploit edited traces to verify the truth of visual content. One group of CNN-based methods focus on the spatial domain to capture artifact traces, such as blending [25, 51],

multiple instance learning [26] patch consistency [59], reconstruction [18, 27], and local mining [9]. The other works transformed images into the frequency domain by applying DCT [43], combining phase spectrum [32], and extracting high-frequency noises [33].

Text-based methods primarily delve into various aspects. Ghanem *et al.* [13] proposes to incorporate topic and affective information extracted from text. Some social context-based methods leverage user feedback [35], news environment [49], propaganda techniques [17] and temporal patterns [16]. Recently, Nan *et al.* [38] and Zhu *et al.* [64] all discover the domain shift issue caused by the word frequency and emotion etc, and propose domain gate and domain memory bank to enrich domain information, respectively.

In contrast to unimodal methods, multimodal methods adhere to incorporate cross-modal features to extract semantic representations [19]. Sabir *et al.* [46] and Wang *et al.* [55] both propose to combine with the external knowledge base to provide complementary semantics information. Qi *et al.* [40] proposes to extract visual entities to understand the high-level semantics of news. Co-attention network [56] and contextual attention network [42] are both designed to better fuse textual and visual features. Moreover, Ying *et al.* [57] proposes improved Multi-gate Mixture-of-Expert networks (iMMoE) to refine and fuse features extracted from multiple views. Ambiguity learning [6] and causal reasoning [7] are separately introduced to address the issue of modal disagreement decisions and spurious correlation in data bias. A recent work [47] presents HAMMER, a powerful model that combines contrastive learning and cross-modal aggregation. However, all these methods are typically trained and deployed within closed systems, overlooking the potential benefits of accessing world knowledge.

### 2.2 Large Vision-Language Models

Large language models (LLMs) such as GPT-3 [3], LLaMA [53] and Vicuna [8], have showcased remarkable performance on various linguistic tasks. More recently, researchers are exploring extending the capability of LLMs to perceive visual signals. LLaVA [31] and Mini-GPT4 [62] first facilitate image-text feature alignment followed by visual instruction tuning. Visual instruction tuning entails additional training of pre-trained models using curated instruction-formatted datasets to enhance models' generalization to unseen tasks. PandaGPT [52] employs a simple linear layer as a bridge between ImageBind [14] and the Vicuna model, allowing for the multimodal input. The success of LVLMs in the general domain has led to the growth of communities such as medical [24], video understanding [21] and image editing [12]. In this work, we leverage the world knowledge inherent in LVLMs for a better understanding of open-world fake news.

### 2.3 Knowledge Augmented Language Models

Utilizing external knowledge to augment language models (LMs) has emerged as a promising solution in knowledge-intensive tasks [22]. One line of works is retrieval-augmented LMs which retrieve relevant passages and incorporate them into LMs. Borgeaud *et al.* [2] proposes a chunked cross-attention module to incorporate the retrieved text as explicit memory. A lightweight retrieval-augmented dual fine-tuning [29] is introduced to retrofit any LLM. Asai *et al.* [1] introduces self-reflection on retrieved passages to

**Figure 2: Architecture of our proposed FKA-Owl, which is built upon the off-the-shelf LVLM consisting of an image encoder and the LLM. Given a manipulated image-text pair, the cross-modal reasoning module (a) first extracts cross-modal semantic embeddings and visual patch features. Then, these visual patch features are processed by the visual-artifact localization module (b) to encode precise artifact embeddings. Finally, the semantic and artifact embeddings are incorporated into the forgery-aware vision-language model (c) combined with image features and the human prompt for deep manipulation reasoning.**

enhance LM's quality and factuality. Nevertheless, the potential of LVLMs augmented with forgery-specific knowledge in multimodal fake news detection remains unexplored.

## 3  PROPOSED METHOD

In this section, we present a unified framework named FKA-Owl which incorporates forgery-specific knowledge into LVLMs for manipulation reasoning. We first introduce the overall framework architecture. Then, we elaborate on our proposed multi-level cross-modal reasoning module, dual-branch visual-artifact localization module, and forgery-aware vision-language model. Finally, we detail the loss function for training.

### 3.1  The Overall Framework

The comprehensive architecture of FKA-Owl is illustrated in Fig. 2. FKA-Owl consists of an image encoder (ImageBind [14]), a cross-modal reasoning module, a visual-artifact localization module, and a large language model (Vicuna [8]). Given a manipulated image-text pair $(I, T)$, we design two lightweight modules to extract semantic correlations and artifact traces as forgery-specific knowledge representations, respectively. Specifically, the cross-modal reasoning module utilizes dual-branch cross-attention mechanisms to guide cross-modal interactions, facilitating the encoding of semantic embeddings. Concurrently, the visual-artifact localization module gathers local spatial information to establish artifact embeddings through supervised localization. Then, the forgery-aware vision-language model leverages both the forgery-specific knowledge and inherent world knowledge for deep manipulation reasoning.

### 3.2  Multi-level Cross-modal Reasoning

The input image and text are encoded first by the frozen pre-trained encoders, denoted as $E_v$ for the image and $E_t$ for the text. Both $E_v$ and $E_t$ originate from the ImageBind, which is aligned with the Vicuna in the off-the-shelf LVLM [52]. To obtain both low-level elements and high-level semantic cues, we partition the image encoder into multiple layers with layer index $l$, enabling the integration of intermediate patch-level features. As shown in Fig. 2-(a), the image features $f_v$ and text features $f_t$ are separately represented as:

$$f_v = \sum_l E_v^l (I), \ f_t = E_t (T). \tag{1}$$

Both features contain information of object instances within a single modality but lack prior insight into objects referenced by the other modality. This absence of complementary information between modalities may hinder cross-modal semantic reasoning. To this end, we devise the dual-branch cross-attention to guide the interaction between visual and textual features, enabling the extraction of semantic correlations. Attention function [54] is performed on normalized query $(Q)$, key $(K)$, and value $(V)$ features as:

$$\text{Attention}(Q, K, V) = \text{Softmax}(\frac{K^T Q}{\sqrt{D}})V. \tag{2}$$

In dual-branch cross-attention, each modal feature (e.g., image) is allowed to serve as queries $Q$, while keys $K$ and values $V$ can be taken from the other modality (e.g., text):

$$u_v = \text{Attention} (f_v, f_t, f_t), \tag{3}$$

$$u_t = \text{Attention} (f_t, f_v, f_v), \tag{4}$$

where $u_v = \{u_v^{\text{cls}}, u_v^{\text{pat}}\}$, $u_t = \{u_t^{\text{cls}}, u_t^{\text{pat}}\}$. Here, $u_v^{\text{cls}}$ and $u_t^{\text{cls}}$ are [CLS] tokens from visual/textual embeddings interacted with text/image information. $u_v^{\text{pat}}$ and $u_t^{\text{pat}}$ are corresponding patch embeddings.

Based on the cross-modal interaction described above, the [CLS] tokens $u_v^{\text{cls}}$ and $u_t^{\text{cls}}$ can naturally focus on the inter-modal semantic correlations. We concatenate these two [CLS] tokens $\{u_v^{\text{cls}}, u_t^{\text{cls}}\}$ as a joint representation of semantic embedding. Then we use a learnable linear layer to project this generated knowledge into the word embedding space of LVLMs, facilitating profound semantic reasoning aided by world factual knowledge.

## 3.3 Dual-branch Visual-artifact Localization

In addition to extracting semantic correlations between visual and textual features, mining artifact traces is also crucial for distinguishing fake news. Unlike [CLS] token, visual patch tokens with position encoding [54] contain richer local spatial information. Given visual patch tokens $u_v^{\text{pat}}$, we propose a visual-artifact localization module to encode them into artifact embeddings, guided by grounding annotations.

As depicted in Fig. 2-(b), the upper branch which comprises a text encoder and a pixel decoder, is designed to utilize language-driven contrastive learning for pixel-wise localization. Dense features with good language alignment can provide complementary benefits for fine-grained segmentation [15, 20]. Specifically, two class prompts are initially encoded by the text encoder to obtain corresponding features $F_p^i$ $(i = 1, 2) \in \mathbb{R}^{2 \times C_{text}}$, representing "natural" and "unnatural" states. Detailed contrastive class prompts are presented in the Supplementary Materials. To restore local spatial details, the pixel decoder with consecutive deconvolution layers converts low-resolution features $u_v^{\text{pat}} \in \mathbb{R}^{hw \times C_{img}}$ into high-resolution features $F_h \in \mathbb{R}^{H \times W \times C_{img}}$. Since patch-level features are not aligned with the textual space, we project both textual features $F_p$ and visual features $F_h$ into a shared representation space. Subsequently, the projected features $\tilde{F}_p \in \mathbb{R}^{2 \times C}$ and $\tilde{F}_h \in \mathbb{R}^{H \times W \times C}$ are employed to compute similarity scores $w \in \mathbb{R}^{H \times W \times 2}$:

$$w^i = \tilde{F}_h \cdot \tilde{F}_p^T, \ (i = 1, 2). \tag{5}$$

By applying the scores $w$ spatially, we can establish the manipulated segmentation map $M_s \in \mathbb{R}^{H \times W \times 2}$ to achieve per-pixel prediction:

$$M_s^j = \text{softmax}(w) = \log\left(\frac{e^{w^j}}{\sum_{i=1}^{2} e^{w^i}}\right), \ (j = 1, 2). \tag{6}$$

To enrich the representation of artifact traces with multiple levels of details, the lower branch employs multi-head attention for patch-level localization. We utilize a learnable token $q_{\text{tok}} \in \mathbb{R}^{1 \times C_{img}}$ serving as a query, while visual features $u_v^{\text{pat}}$ act as both the key and value. Through the multi-head attention mechanism, local information related to artifacts is aggregated within $u_{\text{agg}} \in \mathbb{R}^{1 \times C_{img}}$ under the supervision of bounding box grounding:

$$u_{\text{agg}} = \text{Attention}\left(q_{\text{tok}}, \tilde{u}_v^{\text{pat}}, \tilde{u}_v^{\text{pat}}\right). \tag{7}$$

To leverage the artifact knowledge contained in the two-branch features (i.e., the manipulated segmentation map $M_s$ and the aggregated token $u_{\text{agg}}$), we separately devise multiple convolution layers and a simple linear layer as projectors. In this manner, both $M_s$

and $u_{\text{agg}}$ are converted into continuous artifact embeddings aligned with the final vision-language model.

## 3.4 Forgery-aware Vision Language Model

With the extraction of two types of forgery-specific knowledge, merging this external knowledge into LVLMs becomes imperative. Both the obtained semantic embeddings and artifact embeddings have been refined to lie in the embedding space for compatibility with language models. Moreover, the introduction of MFND instruction data along with answer heuristics and soft prompts can further activate the capacity of LVLMs.

Due to the lack of instruction-follow data in the MFND task, we carefully design an instruction template following the conversational format of the Vicuna model [8], as shown below:

*###Human: <ImageFeature></Img><ForgeryFeature>[Human Prompt] ###Assistant:*

In this prompt, <ImageFeature> represents the visual tokens produced by the image encoder and <ForgeryFeature> is the combination of semantic and artifact embeddings. The human prompt adopts the multiple choice question answering format, as shown in the dialog box of Fig. 2-(c).

To unleash the potential knowledge of LVLMs in solving MFND tasks, we devise two prompt strategies to serve as more informative inputs in Fig. 2-(c). On the one hand, we utilize candidate answer heuristics [45, 48] to present both the question and answer options to LVLMs and make them predict the symbol (e.g., "A") associated with the selected answer. This approach enables the language models to explicitly compare different candidate answers showcasing more accurate responses. On the other hand, we implement soft prompt tuning to introduce learnable continuous vectors while freezing the language models. These vectors combined with semantic embeddings facilitate the extraction of additional semantic information. Meanwhile, this approach reduces the burden of LVLMs to learn forgery-aware alignment, thereby mitigating the catastrophic forgetting problem.

## 3.5 Loss Function

Two groups of loss functions are employed in our training procedure: cross-entropy loss, and two-branch localization losses.

*3.5.1 Cross-entropy Loss.* In the training of language models, cross-entropy loss is employed to measure the disparity between the text sequence predicted by the models and the target text sequence. The formula is computed according to:

$$\mathcal{L}_{\text{ce}} = -\sum_{i=1}^{n} y_i \log(p_i), \tag{8}$$

where $n$ denotes the token count, $y_i$ is the true label for token $i$ and $p_i$ is the corresponding predicted probability.

*3.5.2 Dual-branch Localization Loss.* Two-branch localization losses are designed for the precise encoding of artifact traces guided by pixel-wise and patch-level localization, respectively. Pixel-wise localization introduces focal loss [28] and dice loss [34] to enable supervision for the manipulated segmentation map $M_s$: $\mathcal{L}_{\text{pixel}} = \mathcal{L}_{\text{focal}} + \mathcal{L}_{\text{dice}}$. Patch-level localization involves regressing

the final bounding box with the aggregated token $u_{agg}$ and computing the regression losses with the ground-truth box by introducing L1 loss and GIoU loss [44]: $\mathcal{L}_{patch} = \mathcal{L}_{L_1} + \mathcal{L}_{giou}$. More details about localization losses are provided in the Supplementary Materials.

Finally, the overall loss function is defined as:

$$\mathcal{L} = \mathcal{L}_{ce} + \mathcal{L}_{pixel} + \mathcal{L}_{patch}. \tag{9}$$

## 4 EXPERIMENTS

In this section, we first introduce the overall experimental setup and then provide comprehensive experimental results to demonstrate the superiority of our proposed method.

**Table 1: The statistics of the four subsets within the DGM$^4$ dataset categorized by the news sources.**

| Domain | | BBC | Guardian | USA | Wash. |
|---|---|---|---|---|---|
| **Train** | # Real | 20436 | 55459 | 15472 | 12725 |
| | # Fake | 20375 | 54244 | 16339 | 13134 |
| | Total | 40811 | 109703 | 31811 | 25859 |
| **Test** | # Real | 3156 | 9109 | 2533 | 2078 |
| | # Fake | 6214 | 17919 | 5393 | 4303 |
| | Total | 9370 | 27028 | 7926 | 6381 |

### 4.1 Experimental Setup

*4.1.1 Dataset.* We evaluate the proposed method on DGM$^4$ dataset[1] [47] and Fakeddit dataset[2] [37].

**DGM$^4$.** DGM$^4$ dataset is the recently released large-scale multimodal manipulation dataset which comprises 230K image-text paired samples with over 77K pristine pairs and 152K manipulated pairs. In the DGM$^4$ dataset, image manipulation involves face swapping and facial emotion editing while text manipulation includes sentence replacement and textual sentiment editing. The construction of the DGM$^4$ dataset is based on the VisualNews dataset [30], which is collected from multiple news agencies. Different agencies cover diverse regional perspectives, thematic focus, and language styles (see Supplementary Materials for the analysis of word clouds), resulting in substantial distribution discrepancies. To simulate the open-world domain-shift scenarios, we partition the DGM$^4$ dataset into four subsets based on news sources: BBC, The Guardian, USA TODAY (USA), and The Washington Post (Wash.). The statistics of four subsets are listed in Table 1.

**Fakeddit.** Fakeddit dataset is curated from multiple subreddits of the Reddit platform where data varies in its content, ranging from political news stories to simple everyday posts. We follow the official dataset partition to only use multimodal samples and use the 2-way categorization for this dataset. Furthermore, we preprocess the data by removing excessively short tweets based on their text length. Short texts often lack sufficient information for semantic

[1]https://github.com/rshaojimmy/MultiModal-DeepFake
[2]https://github.com/entitize/Fakeddit

inconsistency detection. For our task, we only use the test set data for the cross-dataset evaluation.

*4.1.2 Evaluation Metrics.* We treat the multimodal fake news detection problem as a binary classification task. Following previous works [39, 47], we apply the Area Under the Receiver Operating Characteristic Curve (AUC), Equal Error Rate (EER), the Accuracy Score (ACC) as our evaluation metrics.

*4.1.3 Baselines.* The proposed FKA-Owl is compared with the following strong baseline models: **1) PandaGPT** [52]: The off-the-shelf PandaGPT model effectively aligns visual features with the text space of LLMs, enabling it to perform complex multimodal tasks in a zero-shot manner. **2) PandaGPT+SPT:** This model integrates PandaGPT with soft prompt tuning [23] by using the predefined question prompt along with learnable continuous vectors to fine-tune the LVLM during the instruction tuning phase. **3) HAMMER** [47]: HAMMER employs two unimodal encoders to encode image and text embeddings with contrastive learning for alignment, and then summarize multimodal information through the multimodal aggregator.

*4.1.4 Implementation Details.* We utilize the visual encoder and text encoder sourced from ImageBind-Huge [14] as the backbone for our image and text feature extractors. Moreover, we employ Vicuna-7B [8] as the inferential LLM, connected through a linear layer. The model is initialized from the instruction-tuned checkpoint provided by PandaGPT. All training images are resized to 224 × 224 and subjected to random horizontal flipping, as well as random perturbation techniques such as JPEG compression and Gaussian Blur following [47]. The multi-level cross-modal reasoning module extracts intermediate patch features from the 8th, 16th, 24th, and 32nd layers of the image encoder. We set the learning rate as 1.5e-5 with a batch size of 16 and a maximum of 10 epochs when trained on the BBC subset. Linear warm-up and the one-cycle cosine learning schedule are adopted. All experiments are conducted on four NVIDIA GeForce 3090 GPUs with PyTorch. More details about hyperparameter settings are provided in the Supplementary Materials.

### 4.2 Performance Comparison

We evaluate the cross-domain performance of our FKA-Owl with baselines in single-domain, multiple-domain, and cross-dataset settings respectively.

*4.2.1 Single-domain Setting.* Table 2 presents the performance of our method and other baseline models in the challenging scenario where a single domain is available. We randomly select one subset of the DGM$^4$ dataset as the source domain for training and the remaining subsets as the target domains for testing. From the results, we make the following observations:

- The performance of the existing MFND method drops significantly when tested on the unknown subsets, which verifies the existence of domain shift caused by the deviation in propagation contents.
- FKA-Owl yields substantial improvement on recent LVLMs, PandaGPT, and PandaGPT with soft prompt tuning, in both intra-domain and cross-domain testing. Such huge improvement

**Table 2: Single-domain performance (%) comparison of baseline models on DGM$^4$ dataset. Specifically, we use one subset for training and the remaining subsets for testing. SPT denotes the utilization of soft prompt tuning. (●) indicates the intra-domain performance. The better results in each group are in boldface.**

| Train | Method | Test | | | | | | | | | | | |
|---|---|---|---|---|---|---|---|---|---|---|---|---|---|
| | | BBC | | | Guardian | | | USA | | | Wash. | | |
| | | AUC↑ | EER↓ | ACC↑ | AUC↑ | EER↓ | ACC↑ | AUC↑ | EER↓ | ACC↑ | AUC↑ | EER↓ | ACC↑ |
| | PandaGPT | 49.99 | 50.06 | 66.31 | 49.58 | 50.19 | 66.30 | 49.47 | 50.37 | 68.04 | 49.51 | 50.47 | 67.43 |
| BBC | PandaGPT+SPT | 54.93 | 47.01 | 48.29 | 53.89 | 47.23 | 54.33 | 51.19 | 49.72 | 55.96 | 52.43 | 48.07 | 54.91 |
| | HAMMER | 87.35 | 21.40 | 79.98 | 80.82 | 26.81 | 76.46 | 65.16 | 39.40 | 69.02 | 67.01 | 37.92 | 68.28 |
| | **FKA-Owl** | **89.61** | **18.61** | **81.55** | **84.95** | **23.55** | **77.08** | **73.10** | **33.91** | **70.50** | **74.81** | **32.29** | **71.52** |
| Guardian | PandaGPT+SPT | 54.66 | 46.91 | 46.04 | 56.57 | 45.21 | 50.37 | 55.99 | 45.85 | 50.41 | 55.58 | 46.11 | 50.15 |
| | HAMMER | 73.74 | 31.53 | 66.74 | 93.90 | **13.38** | **87.45** | 65.34 | 39.61 | 69.23 | 63.24 | 41.17 | 68.50 |
| | **FKA-Owl** | **82.65** | **25.11** | **74.92** | **93.93** | **13.38** | 86.60 | **74.32** | **32.65** | **71.06** | **73.15** | **33.13** | **70.16** |
| USA | PandaGPT+SPT | 50.01 | 50.31 | 59.12 | 52.88 | 47.94 | 59.49 | 56.50 | 45.29 | 58.98 | 53.89 | 47.76 | 60.27 |
| | HAMMER | 68.44 | 35.32 | 69.81 | 74.71 | 30.46 | 74.35 | 85.11 | 22.68 | 79.08 | 81.60 | 25.17 | 76.90 |
| | **FKA-Owl** | **74.17** | **31.23** | **72.91** | **78.82** | **27.63** | **76.66** | **89.64** | **18.69** | **80.96** | **87.76** | **20.25** | **80.68** |
| Wash. | PandaGPT+SPT | 51.22 | 49.50 | 53.43 | 53.03 | 47.43 | 55.15 | 54.67 | 47.05 | 54.88 | 53.93 | 47.09 | 56.43 |
| | HAMMER | 71.29 | 33.54 | 70.59 | 76.78 | 29.40 | 74.21 | 82.11 | 25.66 | 77.35 | 83.30 | 24.26 | 77.64 |
| | **FKA-Owl** | **78.56** | **28.52** | **73.47** | **81.97** | **25.31** | **76.29** | **87.07** | **21.19** | **79.06** | **87.94** | **19.81** | **80.16** |

**Table 3: Multiple-domain performance (%) comparison of baseline models on DGM$^4$ dataset. Specifically, we use two subsets from the identical country for training and the remaining subsets for testing. SPT denotes the utilization of soft prompt tuning. The better results in each group are in boldface.**

| Method | BBC & Guardian | | | | | | USA & Wash. | | | | | |
|---|---|---|---|---|---|---|---|---|---|---|---|---|
| | USA | | | Wash. | | | BBC | | | Guardian | | |
| | AUC↑ | EER↓ | ACC↑ | AUC↑ | EER↓ | ACC↑ | AUC↑ | EER↓ | ACC↑ | AUC↑ | EER↓ | ACC↑ |
| PandaGPT+SPT | 50.59 | 49.56 | 54.44 | 51.95 | 48.68 | 52.55 | 52.38 | 48.09 | 52.54 | 51.74 | 48.75 | 53.13 |
| HAMMER | 63.45 | 39.81 | 69.19 | 62.59 | 40.34 | 68.64 | 73.95 | 32.48 | 71.86 | 80.04 | 27.54 | 76.06 |
| **FKA-Owl** | **75.17** | **33.35** | **71.42** | **75.15** | **32.85** | **70.52** | **81.06** | **26.21** | **75.26** | **85.88** | **22.41** | **78.87** |

demonstrates the effectiveness of forgery-specific knowledge augmentation in our framework.

• Compared with the state-of-the-art method, HAMMER, our approach shows more remarkable improvement in cross-domain testing. For instance, for models trained on the BBC subset, FKA-Owl achieves a 7.7% increase in AUC when testing on the Washington Post subset. This may be credited to the effective utilization of inherent world knowledge from LVLMs in mitigating distribution discrepancies. The combination of forgery-specific knowledge and world knowledge facilitates profound manipulation reasoning in FKA-Owl.

*4.2.2 Multiple-domain Setting.* The inclusion of domestic news such as the BBC and The Guardian from British, as well as USA TODAY and The Washington Post from America, increases dataset diversity in practical scenarios. We select two subsets from identical countries for training and the remaining two for testing. The results are summarized in Table 3. Our FKA-Owl exhibits significant superiority over both PandaGPT using soft prompt tuning and HAMMER by a large margin. This reveals the effectiveness of our framework in instance-wise domain generalization guided by world knowledge derived from LVLMs, even when jointly learning multiple source domains.

**Table 4: Cross-dataset performance (%) comparison of baseline models on the Fakeddit dataset when trained on DGM$^4$ dataset. Specifically, we use one subset from the DGM$^4$ dataset for training and the Fakeddit dataset for testing. The better results in each group are in boldface.**

| Test | Method | Train | | | | | | | | | | | |
|---|---|---|---|---|---|---|---|---|---|---|---|---|---|
| | | BBC | | | Guardian | | | Usa Today | | | Washington Post | | |
| | | AUC↑ | EER↓ | ACC↑ | AUC↑ | EER↓ | ACC↑ | AUC↑ | EER↓ | ACC↑ | AUC↑ | EER↓ | ACC↑ |
| Fakeddit | HAMMER | 35.61 | 60.68 | 40.45 | 44.81 | 53.72 | 44.82 | 36.09 | 60.58 | 39.80 | 35.76 | 61.14 | 38.55 |
| | **FKA-Owl** | **44.96** | **53.54** | **46.44** | **57.17** | **44.13** | **55.47** | **43.07** | **55.30** | **43.53** | **40.20** | **57.33** | **43.31** |

**Table 5: Ablation study of component modules. We evaluate the AUC (in %), EER (in %), and ACC (in %) of variant models on the remaining three subsets when trained on the BBC subset. ML: the extraction of multi-level patch features in the cross-modal reasoning (CR) module. DB: the extraction of dual-branch artifact features in the visual-artifact (VA) Localization module. Avg. denotes the mean value on the three testing subsets.**

| Componet Module | | Test | | | | | | | | | | | |
|---|---|---|---|---|---|---|---|---|---|---|---|---|---|
| ML&CM Reasoning | DB&VA Localization | Guardian | | | USA | | | Wash. | | | Avg. | | |
| | | AUC↑ | EER↓ | ACC↑ | AUC↑ | EER↓ | ACC↑ | AUC↑ | EER↓ | ACC↑ | AUC↑ | EER↓ | ACC↑ |
| | | 53.89 | 47.23 | 54.33 | 51.19 | 49.72 | 55.96 | 52.43 | 48.07 | 54.91 | 52.50 | 48.34 | 55.07 |
| ✓ | | 83.53 | 23.95 | 77.03 | 67.64 | 36.45 | 69.52 | 70.97 | 34.60 | 70.33 | 74.05 | 31.67 | 72.29 |
| | ✓ | 81.79 | 26.76 | 73.98 | 66.59 | 39.02 | 65.96 | 69.60 | 36.67 | 67.25 | 72.66 | 34.15 | 69.06 |
| ✓ | ✓ | **84.95** | **23.55** | **77.08** | **73.10** | **33.91** | **70.50** | **74.81** | **32.29** | **71.52** | **77.62** | **29.92** | **73.03** |
| **FKA-Owl** w/o ML | | 81.27 | 26.70 | 75.49 | 65.25 | 39.71 | 69.20 | 68.97 | 36.54 | 69.21 | 71.83 | 34.32 | 71.30 |
| w/o DB (upper) | | 83.70 | 23.92 | 77.29 | 62.85 | 40.05 | 70.00 | 67.73 | 36.27 | 70.49 | 71.43 | 33.41 | 72.59 |
| w/o DB (lower) | | 84.12 | 23.88 | 76.74 | 70.37 | 36.36 | 69.66 | 73.50 | 32.87 | 70.30 | 76.00 | 31.04 | 72.23 |

*4.2.3 Cross-dataset Setting.* To comprehensively represent the contextual diversity inherent in multimodal fake news detection tasks, We select one subset from DGM$^4$ for training and evaluate the performance on the Fakeddit dataset. As shown in Table 4, the performance of all methods notably decreases when tested on the Fakeddit dataset, which implies that the difference in the distribution of different datasets does exist. Furthermore, we can observe that our method outperforms the state-of-the-art model HAMMER when trained on different subsets, confirming the generalizability of our approach.

## 4.3 Ablation Study

We perform several ablation experiments to explore the necessity of the proposed component modules, prompt strategies, and LVLMs knowledge respectively, and analysis of potential module choices. In the following experiments, all ablation results are evaluated on the remaining three subsets when trained on the BBC subset.

**Table 6: Ablation study of prompt strategies. AUC (in %) of variant models is reported on the remaining three subsets when trained on the BBC subset. SPT denotes soft prompt tuning, whereas CAH refers to candidate answer heuristics.**

| Strategy | Guardian | USA | Wash. | Avg. |
|---|---|---|---|---|
| w/o SPT | 84.76 | 71.98 | 74.29 | 77.01 |
| w/o CAH | 83.14 | 67.38 | 70.55 | 73.69 |
| w/o SPT & CAH | 83.32 | 66.12 | 69.94 | 73.13 |
| **FKA-Owl** | **84.95** | **73.10** | **74.81** | **77.62** |

*4.3.1 The Effect of the Component Modules.* In Table 5, we conduct a comprehensive ablation study on the proposed component modules to verify their effectiveness. The first row of Table 5 shows our baseline model that only performs soft prompt tuning, achieving an

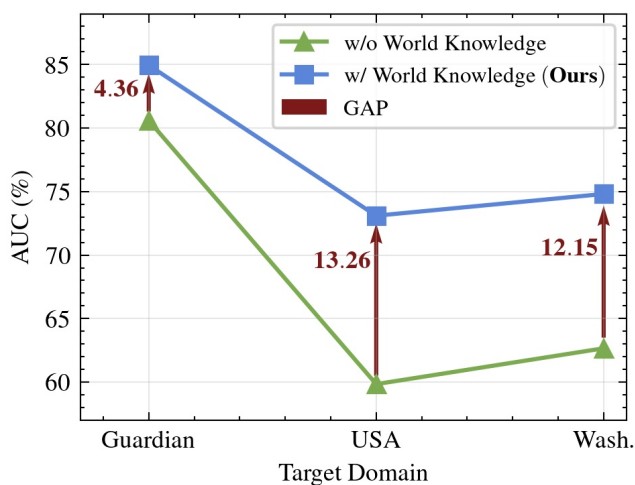

**Figure 3: Ablation study of the world knowledge inherent in large vision-language models.**

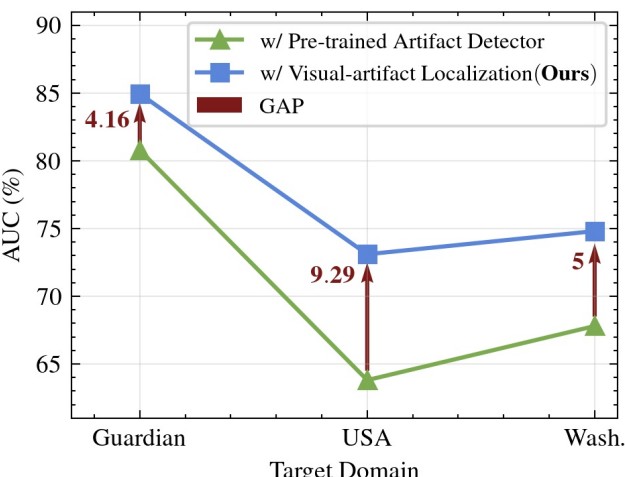

**Figure 4: Ablation study of the potential module choice of using pre-trained artifact detector to replace visual-artifact localization module.**

average AUC of 52.5%. Based on this baseline, we further introduce two separate modules: a multi-level cross-modal (ML&CM) reasoning module and a dual-branch visual-artifact (DB&VA) localization module, with 21.55% and 20.16% improvement in average AUC, respectively. This implies that forgery-specific knowledge augmentation is indispensable for our framework. Comparatively, our model with complete two modules obtains the best performance increasing by 25.12%, indicating the effectiveness and complementarity of these two modules. Moreover, we test the performance of FKA-Owl removing the multi-level features (w/o ML), and FKA-Owl removing any one of the dual-branch features (w/o DB). These variant models lack rich features to represent forgery-specific knowledge leading to a great decrease in cross-domain performance.

*4.3.2 The Effect of the Prompt Strategies.* The prompt strategies designed in the alignment process and the corresponding results for each case are tabulated in Table 6. First, when removing continuous prompt vectors (w/o SPT), the performance drops a little bit. In addition, after removing the candidate answer options (w/o CAH) in the human instruction prompt, the average performance decreases from 77.62% to 73.69%. In particular, the third row of Table 6 represents that none of both strategies is employed in our proposed framework. Our method substantially outperforms this variant model, implying both two strategies enable the introduction of implicit information to fully activate the capacity of LVLMs.

*4.3.3 The Effect of the LVLMs Knowledge.* To analyze the impact of world knowledge derived from LVLMs, we compare our method with the common practice of processing fused embeddings in a supervised classification manner [47]. This variant model (w/o World Knowledge) replaces the Vicuna model in FKA-Owl with a binary classifier to predict true/fake labels. As shown in Fig. 3, harnessing the inherent knowledge in LVLMs improves an average performance by 9.92 points over the variant model. Furthermore, for target domains from distinct countries exhibiting huge distribution differences, FKA-Owl yields more significant improvements in

13.26% and 12.15%. This could be attributed to the fact that world knowledge from LVLMs can effectively guide the representation of agnostic instances.

*4.3.4 Analysis on Potential Module Choice.* We replace the artifact extractor module in our framework for other design choices. This variant model (w/ Pre-trained Artifact Detector) replaces the visual-artifact localization module in FKA-Owl with the off-the-shelf pre-trained detector [47] to obtain the artifact embeddings. The pre-trained detector employs the VIT backbone with supervised training by grounding annotations. As depicted in Fig. 3, our visual-artifact localization module brings a significant improvement over the variant model. This could be attributed to the fact that Our designed module leverages the intrinsic visual encoder in LVLMs to extract artifact traces, thereby alleviating the burden of aligning forgery-specific knowledge with LVLMs.

## 5 CONCLUSION

Our work presents FKA-Owl, a novel framework that leverages rich world knowledge from LVLMs and enhances them with forgery-specific knowledge, to tackle the domain shift issue in multimodal fake news detection. Two types of critical forgery-specific knowledge are augmented in FKA-Owl: semantic correlation between text and images and artifact trace in image manipulation. To inject this knowledge into the LVLM, we first propose two lightweight specialized modules to learn their representations respectively. Then, we transform the generated knowledge into refined embeddings for alignment with language space. The candidate answer heuristics and soft prompts are introduced as supplementary inputs to unleash the extensive knowledge of LVLMs. Extensive experiments verify that FKA-Owl shows superior cross-domain performance compared to the state-of-the-art methods.

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
