# OpenReview forum: "FKA-Owl: Advancing Multimodal Fake News Detection through Knowledge-Augmented LVLMs"
_acmmm.org/ACMMM/2024/Conference — MM2024 Poster_

### Official Review · Reviewer_k3vA · 2024-05-14

**Rating:** 3
**Confidence:** 3

**Summary:**

The paper aims at leveraging forgery-specific knowledge to augment LVLMs for combating fake news.

**Strengths:**

The proposed method focuses on cross-modal semantic correlation and artifact traces of image manipulation. Authors conduct some experiments for validation. The proposed method achieves good cross-domain performance.

**Limitations:**

- The author propose to leverage artifact-related knowledge to augment LVLMs for fake news detection. However, the chosen validation datasets are not real-world fake news datasets, which contain fake news that are spreading or have spread. The DGM4 dataset construct its false samples by intentionally applying forgery methods. Fakeddit collects false samples from subreddits include “photoshopbattles submission”, “fakealbumcovers”, “photoshopbattles”, and so on. Both datasets contain deliberately forged samples, which makes them suitable for multi-modal manipulation detection, but not appropriate for testing fake news detection ability.

- The cross-domain ability of FKA-Owl may stem from the common forgery methods shared by the dataset but not by the cross-domain reasoning ability. What is the cross-domain performance without visual-artifact localization?

**Suitability:**

2

---

### Official Review · Reviewer_L7Hg · 2024-05-24

**Rating:** 4
**Confidence:** 3

**Summary:**

The paper leverage Large Vision-Language Models (LVLMs) to tackle the domain shift issue in multimodal fake news detection. Besides, the paper utilizes multi-modal semantic discrepancies and image artifact tracing to augment LVLMs. The experimental results indicate that the proposed method achieves good cross-domain detection performance.

**Strengths:**

1. The explore of LVLMs for fake news detection is interesting and worthy of investigation.
2. To address the limitations of directly applying LVLMs, this article introduces the forgery-specific knowledge to enhance LVLMs. And ablation experiments demonstrate the effectiveness of the proposed method.

**Limitations:**

1. The proposed method is based on two assumptions, 1) the inconsistency between the text and image, and 2) a local part of the image is tempered. It is not clear what will happen when the senmentic of the image and the text are consistent or the image is generated by AI techniques.
2. The techniqucal contribution seems limited, where most of the components are borrowed from the existing techniques.
3. It is not clear how the instruction template would affect the performance.

**Suitability:**

3

---

### Official Review · Reviewer_d5HW · 2024-05-27

**Rating:** 4
**Confidence:** 3

**Summary:**

This paper presents a novel framework called FKA-Owl to address the challenge of detecting multimodal fake news that combines both text and images. The proposed method leverages Large Vision-Language Models (LVLMs) and enhances them with forgery-specific knowledge to better detect manipulations. FKA-Owl incorporates two types of forgery-specific knowledge: semantic correlation between text and images and artifact traces in image manipulation. Two specialized modules are designed to extract these features and inject them into LVLMs. The framework demonstrates superior cross-domain performance in detecting fake news compared to existing methods.

**Strengths:**

1）By integrating forgery-specific knowledge, FKA-Owl significantly improves the ability of LVLMs to detect subtle manipulations in multimodal content that traditional models might miss.
2）The introduction of specialized modules for encoding semantic correlations and artifact traces is particularly innovative, enhancing the model's capability to interpret complex multimodal data.

**Limitations:**

Add case study to demonstrate the role of Forgery specific knowledge in detecting fake news.

FKA-Owl represents a significant advancement in the field of multimodal fake news detection. By combining the strengths of LVLMs with forgery-specific knowledge, the framework addresses critical challenges related to domain shift and cross-modal manipulations. The comprehensive evaluation and ablation studies provide strong evidence of its effectiveness.

**Suitability:**

2

---

### Meta-Review · Area_Chair_9Rj7 · 2024-07-11

**Recommendation:** Accept (Poster)
**Confidence:** 3

**Metareview:**

All the reviewers suggest to accept this submission, giving final rating as weak accept.